# Thermal Oxidative Stability of Biodiesel/Petrodiesel Blends by Pressurized Differential Scanning Calorimetry and Its Calculated Cetane Index

**Jilliano B. Silva [1], Josue S. Almeida [2], Rodrigo V. Barbosa [3], Glauber J. T. Fernandes [4], Ana C. F. Coriolano [5], Valter J. Fernandes, Jr. [4],\* and Antonio S. Araujo [5],\***

[1] Postgraduate Program in Petroleum Science and Engineering, Federal University of Rio Grande do Norte, Natal 59078-970, Brazil; jilliano.lcl@gmail.com

[2] Postgraduate Program in Materials Science and Engineering, Federal University of Rio Grande do Norte, Natal 59078-970, Brazil; santiago.lcl@gmail.com

[3] Postgraduate Program in Chemistry, Federal University of Rio Grande do Norte, Natal 59078-970, Brazil; rodrigovictor62@gmail.com

[4] Laboratory of Fuels and Lubricants, Institute of Chemistry, Federal University of Rio Grande do Norte, Natal 59078-970, Brazil; glauberturolla@gmail.com

[5] Laboratory of Catalysis and Petrochemistty, Institute of Chemistry, Federal University of Rio Grande do Norte, Natal 59078-970, Brazil; catarina.coriolano@gmail.com

\* Correspondence: valter@quimica.ufrn.br (V.J.F.J.); araujo.ufrn@quimica.ufrn.br (A.S.A.)

**Abstract:** Diesel fuel mixtures with high concentrations of biodiesel have been investigated to analyze the technical feasibility of their use in diesel cycle engines regarding thermal and oxidative properties. The results of combined techniques of oxidative stability, high Pressurized Differential Scanning Calorimetry (P-DSC), Calculated Cetane Index (CCI), and calorific power were used to verify the effect of the thermal-oxidative stability as a function of the percentage of biodiesel in the mixtures. The obtained results evidenced that the thermal and oxidative stability decreased with the addition of biodiesel from 50 to 5% $v/v$. Low stability fuels require rapid use as the oxidation compounds degrade the product and impair vehicle performance, as well as lead to corrosion and clogging problems in various mechanical systems.

**Keywords:** biodiesel/petrodiesel blends; thermal-oxidative stability; Calculated Cetane Index (CCI)



## 1. Introduction

Due to the environmental problems that have arisen in recent decades, such as air pollution and greenhouse gas emissions, the use of fossil fuels by other renewable energy sources is increasingly being sought. In Brazil, road diesel is the most widely used petroleum-derived fuel, and its annual consumption can be correlated with the economic situation of the period analyzed since the transportation matrix of products produced in the country is based on the road network. This petroleum derivative is defined as a liquid fuel composed of hydrocarbons with 8 to 16 carbon chains and, to a lesser extent, nitrogen, sulfur, and oxygen [1]. It is mainly used in diesel cycle engines (internal combustion and compression ignition) in the road, rail, and marine vehicles, and electric power generators.

According to the National Agency of Petroleum, Natural Gas and Biofuels, Brazil was the seventh country among the largest consumers of petroleum products, with a forecast of 19% increase in national demand between 2016 and 2026. Thus, the use of biofuels should be increased so that the country can meet the targets agreed upon at the United Nations Conference on Climate Change (COP21), held in 2015 in Paris. In this context, biodiesel presents itself as a clean energy source capable of replacing, in whole or in part, diesel oil without any detriment to the performance of vehicles operating on the diesel cycle, since no technical modifications to conventional engines are required for the diesel engine.

Biodiesel is defined as a fuel composed of long-chain alkyl carboxylic acid esters produced from the transesterification and/or esterification of fatty materials of plant or animal origin. This biofuel has several advantages compared to diesel oil, of which one can cite the low sulfur content, being practically exempt, has higher lubricity, higher cetane number and flash point (which reduces the risk of accidents during the storage). However, regarding the physicochemical parameters, biodiesel has low oxidation stability. The presence of unsaturated bonds in the esters of biodiesel molecules favors the oxidative degradation of biofuel when subjected to high temperatures, the presence of metals, and exposure to an oxidizing atmosphere. The higher the concentration of these unsaturated bonds, the easier the fuel will be to oxidize [2].

When undergoing the oxidative process, biodiesel decomposes and generates as its main by-products organic acids. Metals, particularly copper, copper alloys, lead, zinc, and tin, which are used in fuel and supply systems, easily corrode in the presence of these compounds [3]. The effects of acids on a metallic fuel tank are particularly severe. Even if only slight corrosion is formed, the oxidizing organic acids react and form metal salts. These salts which precipitate in the fuel pass through the fuel filter and adhere to some mechanical components, such as the injector pump and the nozzle surfaces, forming deposits. Other immiscible substances formed include polymers, sludge, and oxidation products, which at some point cause blockage of the filters [4]. Oxidation intermediates, peroxides, deteriorate the plastic, and elastomers that are used in piston seal rings degrade at high temperatures [5].

A lot of research has been carried out to understand the thermal properties and oxidative properties of biodiesel [6–10]. To increase the stability of biodiesel, the effects of antioxidant additives have been investigated [11–13]. Biodiesel has also been prepared using supercritical methanol, to increase its oxidative stability [14]. The addition of biodiesel to mineral diesel has been carried out, aiming to increase the stability of biodiesel-petrodiesel blends during storage [15,16]. It has been found that the factors that affect oxidative stability can be determined by assessing the induction period and acidity. Rancimat [17] and Pressurized Differential Scanning Calorimetry (P-DSC) [17,18] have been used as analytical tools for thermal-oxidative properties of biodiesel. It was observed that the induction period at different storage times was dependent on the degree of saturation of fatty acid methyl esters [19]. Thermal-oxidation stability and cold flow property are the main problems associated with the use of biodiesel. Different types of biodiesels, such as palm, soybean, and rapeseed biodiesels had been blended with different weight ratios. The oxidation stability and the cold point of the blended biodiesels presented a relationship with the compositions of the major fatty acid components.

The Calculated Cetane Index (CCI) is a physicochemical parameter of diesel related to the burning of fuel in the engine, which measures the ignition quality and its value directly affects the ignition and the operation with charge [20]. The ignition quality is assessed by measuring the period between injection and the start of fuel combustion. A fuel with a high CCI has an ignition delay and starts to burn right after being injected into the engine [21].

The CCI represents a simple correlation with cetane number, serving in many cases as a substitute due to its simplicity of obtaining. This parameter measures the ignition quality of fuels to be used in diesel cycle engines. According to the literature [22], for heavy fuel oils, the ignition properties are typically ranging from CN = 18.7 to above CN = 40. Fuel ignition quality depends on Cetane Number (CN) for different heavy fuel oils or marine fuel diesel, as given in Table 1.

**Table 1.** Cetane Number (CN) and ignition quality of heavy fuel oil and marine diesel oil [22].

| FIA Cetane Number (CN) | Heavy Fuel Oil | Marine Diesel Oil |
|:---:|:---:|:---:|
| <20 to 25 | Very bad ignition properties | Unfit for use |
| 25 ≤ FIA CN < 28 | Bad ignition properties | Very bad ignition properties |
| 28 ≤ FIA CN < 35 | Acceptable ignition properties | Bad ignition properties |
| 35 ≤ FIA CN < 40 | Good ignition properties | Acceptable ignition properties |
| 40 ≤ FIA CN < 45 | Very good ignition properties | Good ignition properties |
| FIA CN ≥ 45 | Very good ignition properties | Very good ignition properties |

The Calculated Cetane Index (CCI) represents what the fuel ignition delay will be on the diesel engine. The lower the CCI value, the greater the ignition delay. Low values of this parameter result in a large amount of fuel, which remains in the cylinder without burning at the right design time. Among several problems, the main one is reflected in the engine malfunction because when the last fuel portion combustion occurs, the amount of energy generated will be greater than that required for the end of the duty cycle, resulting in abnormal efforts—over the piston, causing mechanical damage and loss of power.

In recent years, important studies have been published regarding the use of biodiesel as a substitute for petroleum fuels. However, few studies discuss the effect of using diesel blends with high concentrations of biodiesel as a way to decrease the dependence on this fossil fuel. In this context, this work aims to evaluate the thermal and oxidative stability of biodiesel/petrodiesel blends containing 5; 10; 15; 20; 30, and 50% $v/v$ of biodiesel.

## 2. Materials and Methods

### 2.1. Preparation of the BX Blends

Biodiesel/Petrodiesel blends were prepared at volumetric concentrations of 5, 10, 15, 20, 30, and 50% vol. The samples were signed as BX, where "X" represents the volume of biodiesel. Thus, we have B5, B10, B15, B20, B30, and B50. All samples were prepared to start from a commercial S10 mineral diesel oil (biodiesel free, with 10ppm of S) and biodiesel from soybean oil, both in accordance with the specifications of the Brazilian National Agency of Petroleum, Natural Gas and Biofuels (ANP). First, the mixtures were prepared with the calculated concentrations in grams per liter of the mixture corresponding to the volumetric concentration of interest. For the conversion of g dm$^3$ concentrations to % $v/v$, the specific biodiesel mass of 883 kg m$^{-3}$ was used as the conversion factor.

### 2.2. Calculated Cetane Index (CCI)

The CCI parameter was determined using Equation (1), according to ASTM D976 [23]. This equation takes into account the specific mass measured at 15 °C and 50% vol of recovered fuel in this temperature. These pieces of information are obtained from the distillation curve or the BX blends, according to ASTM D86. The CCI equation is

$$CCI = 454.74 - 1641.416\,D + 774.74\,D^2 - 0.554\,B + 97.803\,(\log B)^2 \tag{1}$$

where D = Density at 15 °C in g/cm$^3$ and determined by methods D1298 [24] or D4052 [25]; B = Temperature measured for 50% volume of the recoverable fuel, as determined by the ASTM D86 [26] method and corrected for barometric pressure.

### 2.3. Calorific Power

The calorific value is a property that represents the amount of thermal energy present in the fuels, which is determined by the complete combustion of the specified amount at constant pressure under normal conditions. The calorific power of the mixtures was determined according to the methodology described in ASTM D4809 [27] by burning the BX blends in an oxygen pump C5000 model, manufactured by IKA. The heat of combustion, calculated from temperature observations before, during, and after combustion, was obtained considering the thermochemical and heat transfer corrections. The adiabatic method

was adopted for these tests, since this method minimizes heat loss, keeping the ambient temperature equal to the sample temperature, making it more accurate to measure the calorific power of solids or liquids.

### 2.4. Oxidation Stability

Oxidation is an exothermic process, being a thermodynamically irreversible reaction. The oxidative stability is a parameter that predicts the useful life of a given material in relation to its resistance to decomposition in an oxidizing atmosphere. The Oxidative Induction Time (OIT) is defined as the time for the appearance of sample oxidation under a certain temperature. The test determines the time interval between the beginning of the test and the appearance of oxidation of the sample, being signaled by a rapid increase in the heat or temperature of the sample. The OIT is determined by extrapolating the onset temperature in the thermogram.

The oxidative stability was determined using a Rancimat equipment model 843 from Metrohm according to the methodology described in the European standard EN 15751 method [28]. For all samples a mass of 7.50 g $\pm$ 0.01 g was used and submitted to a constant temperature of 110 °C in a 25 mm long reaction vessel and at a moisture-free air flowing at a rate of 10 dm$^3$/h. As the oxidation products form, airflow transports the compounds from the reactor to the measuring vessel, which contains ca. 50 cm$^3$ of distilled water, where their electrical conductivity is constantly monitored. The induction period, defined as the time elapsed between the start of the analysis and the time when oxidation product formation begins to increase rapidly, was determined to be the tipping point on the electrical conductivity curve ($\mu$S cm$^{-1}$) versus time (h) [29,30].

### 2.5. Pressure DSC

Differential Scanning Calorimetry (DSC) provides information about physical and chemical changes that involve endothermic, exothermic processes, or changes in heat capacity. The Pressurized DSC (P-DSC) analysis is used to evaluate the oxidative stability of materials using differential heat flow between the sample and the reference thermocouple under different conditions of temperature and pressure. The main advantages of P-DSC are the use of a small amount of sample, less analysis time, as it operates under high temperature and pressure, accelerating the oxidation reaction.

The P-DSC analysis for the BX blends were carried out in a Netzsch DSZ 204 HP equipment, according to the methodology described in ASTM E2009 [31]. The experimental conditions used by the dynamic method were: approximately 10 mg of sample mass; the initial temperature of 40 °C; linear heating ramp of 20 °C min$^{-1}$ to final temperature 500 °C; the oxidizing atmosphere of $O_2$ flowing at 35 mL min$^{-1}$ pressure of 3.5 MPa. The peaks related to the exothermic oxidation events of the blends were analyzed to establish the initial oxidation temperatures and correlations between peak intensity, peak area, or initial oxidation event temperature versus an increase in biodiesel concentration.

## 3. Results and Discussion

### 3.1. Calculated Cetane Index (CCI)

To calculate the Calculated Cetane Index (CCI), the BX blends were submitted to atmospheric distillation and specific mass tests, and Equation (1) was applied for the calculations. The distillation curves are shown in Figure 1. The data obtained from distillation curves and specific mass for the blends are given in Tables 2 and 3, respectively.

The choice of a specific mass, as one of the variables of this equation, can be justified by the analysis that this parameter is related to the total energy potential of the fuel. The higher the density, the larger the mass of fuel being injected by the nozzle into the cylinder per unit volume. Variations outside the technical specification in density make it impossible for a balanced air/fuel mixture, resulting in energy losses. Regarding distillation temperatures, it can be stated that the 50% *v/v* recovered is associated with the relationship between the content of light and heavy fractions in the product. The characteristics of

this fraction influence the engine warm-up time, allowing uniform operating conditions. Controlling this distillation point contributes to good engine performance when the engine is already in a steady-state and at speed pickups.

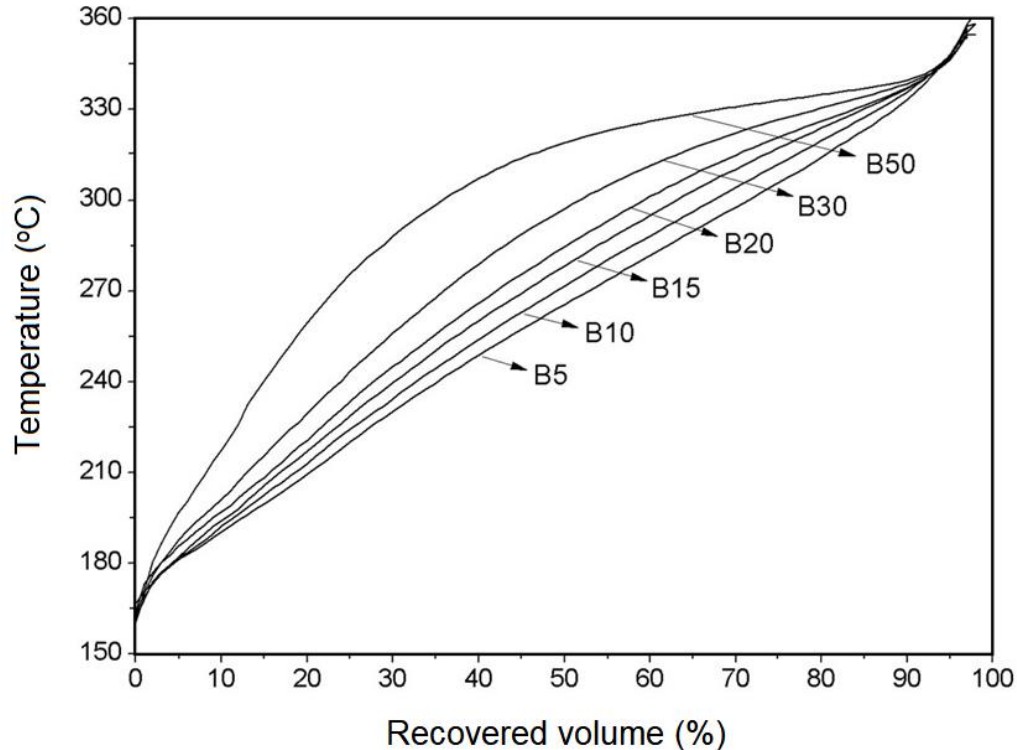

**Figure 1.** Distillation curves were obtained according to ASTM D86 for the biodiesel/petrodiesel blends, showing the profiles for B5, B10, B15, B20, B30, and B50 samples.

**Table 2.** Recovered temperatures for BX biodiesel/petrodiesel blends, according to ASTM D86.

| Blend | Recovered Temperatures (°C) | | |
|---|---|---|---|
| BX | 10% *v/v* | 50% *v/v* | 90% *v/v* |
| B5 | 190.3 | 265.6 | 333.3 |
| B10 | 192.2 | 271.6 | 335.6 |
| B15 | 194.2 | 278.5 | 337.1 |
| B20 | 196.9 | 284.5 | 337.2 |
| B30 | 201.2 | 297.3 | 338.5 |
| B50 | 217.9 | 319.0 | 339.8 |

**Table 3.** Specific mass at 15 °C for BX biodiesel-diesel blends according to ASTM D4052.

| Blend | Specific Mass |
|---|---|
| BX | kg/dm$^3$ (15 °C) |
| B5 | 834.3 |
| B10 | 836.8 |
| B15 | 839.2 |
| B20 | 841.7 |
| B30 | 846.6 |
| B50 | 856.7 |

Although this calculation methodology does not take into account the 10% *v/v* and 90% *v/v* recovered temperatures, as is the case for calculating the cetane number according

to ASTM D4737 [32], these two temperatures also have their importance. The control of the 10% $v/v$ recovered aims to ensure the minimum amount of light fractions that vaporize and burn easily, ensuring the start of vehicle operation (cold start). For the 90% $v/v$ recovered, limiting this temperature aims to minimize the formation of deposits in the combustion chamber and spark plugs. The specification limit must be required to prevent heavy unburnt fractions from leaking into the engine crankcase and contaminating the lubricating oil. With respect to pollutant emissions, heavier hydrocarbons require higher firing temperatures.

From a chemical point of view, the cetane number refers to fuel performance when compared to the performance of a mixture of n-hexadecane with α-methyl naphthalene. n-hexadecane, paraffinic chemical formula $C_{16}H_{34}$, is assigned a cetane number of 100, and α-methyl-naphthalene, aromatic chemical formula $C_{10}H_7CH_3$, is assigned a cetane number of zero. Fuels with a high content of paraffin compounds have a high cetane number while compounds rich in aromatic hydrocarbons have a low cetane number. Low CCI values lead to difficulty in cold starting, piston deposition, and engine malfunction, and increase the emission of pollutants such as hydrocarbons, carbon monoxide, and particulates [33].

For the analyzed mixtures, a gradual increase in the calculated cetane index was observed between the samples from B5 to B30, and thereafter a decrease was observed for the sample B50, in which its CCI value returned approximately to that found in the B20 sample, as given in Table 4.

**Table 4.** Calculated Cetane Index (CCI) values for BX blends.

| Blend | CCI |
|:---:|:---:|
| **BX** | **(Dimensionless)** |
| B5 | 52.2 |
| B10 | 52.6 |
| B15 | 53.2 |
| B20 | 53.4 |
| B30 | 53.9 |
| B50 | 53.5 |

This behavior can be attributed to the fact that the cetane number decreases with the increasing of the unsaturated bonds that are present in biodiesel linoleic acid and/or esters, and increases with the presence of saturated esters [34]. As the concentration of biodiesel in the blends increased, the calculated cetane index gradually increased until the presence of unsaturation bonds. However, the BX blends presented values in accordance with the specification described in ASTM D7467 [35], which establishes the minimum value of 40.

### 3.2. Calorific Power

The calorific power values for the BX blends were determined after weighing approximately 0.5 g of each sample and then inserted into the calorimeter combustion vessel under controlled conditions. With the aid of cotton yarn of known calorific value, the mixture was ignited, and then the temperature variation was measured with the aid of a reading instrument that gives the method accuracy. The heat produced by the combustion, the unit quantity of the fuel, after burning at constant volume in the calorimetric pump, were determined. The combustion of the sample was considered complete when all material resulting from the combustion became liquid water and gases.

This behavior can be attributed to the fact that the cetane number, which in this study was correlated to the calculated cetane index, decreases with the increase in unsaturated bonds that are present in biodiesel linoleic acid and/or esters and increases with the presence of saturated esters [10]. As the concentration of biodiesel in the blends increased, the calculated cetane index gradually increased until at some point the presence of unsaturation caused the value of this parameter to decrease. However, both blends have

shown acceptable values in accordance with the specification described in ASTM D7467, which establishes the minimum value of 40.

The calorific value of the BX samples, defined as the energy released during the burning of a fuel per unit mass or volume, is one of the best energy efficiency indicators to verify the technical applicability of its use in the conversion of thermal energy into mechanical, as is the case for internal combustion engines. Hydrogen has the highest energy per unit mass compared to any fuel as it is the lightest element and has no low carbon calorific atoms from hydrocarbons. Therefore, it can be stated that the higher the percentage of hydrogen and the lower the percentage of carbon in the fuel, the greater the value of its calorific value. Quantitatively, the calorific value can range from 31.4 kJ/g for premium quality coal to 141.8 kJ/g for hydrogen [36].

In calculating the calorific value, measured by the heat released during combustion, the temperatures of the reactants shall be considered equal to the temperatures of the combustion products. Water present in combustion products may be in both liquid and gaseous forms. If present in the liquid state, the energy released in the process will refer to the higher calorific value, and if present in the gaseous state, will refer to the lower calorific value. The difference between the lower calorific value—PCI (steam) and the higher calorific value—PCS (condensed water) will be the amount of energy relative to the latent heat of water vaporization. The difference between these two values will be greater the higher the percentage of hydrogen in the fuel 21. In all situations, the value of PCS is higher than the value of PCI because it accounts for the condensation enthalpy of water, except in cases where the fuel in question does not have hydrogen in its composition because there will be no formation of water; In these cases, the PCS and PCI values are the same. For the mixtures under study, the higher calorific value was analyzed.

In Table 5 is given the higher calorific power obtained for the BX samples. For the blends analyzed, there was a decrease in calorific value with the increase in biodiesel concentration. However, there was only a maximum decrease of only 3.8% between blend B5 and blend B50, which in terms of energy availability will not cause major damage to the use of blends with high concentrations of biodiesel in diesel oil.

**Table 5.** Higher Calorific Power (HCP) for the BX biodiesel-diesel samples.

| Blend | Higher Calorific Power |
|:---:|:---:|
| **BX** | **(kJ/g)** |
| B5 | 45.56 |
| B10 | 45.27 |
| B15 | 44.98 |
| B20 | 45.14 |
| B30 | 44.89 |
| B50 | 43.85 |

It is also noteworthy that the calorific value of biodiesel is very close to mineral diesel oil, with diesel being only 5% $v/v$ higher than biodiesel. However, as the combustion of biodiesel is more complete and therefore more efficient, this biofuel will have a specific consumption equivalent to petroleum derivatives. In addition, it is known that increasing the ester chain may increase the calorific value of biodiesel.

### 3.3. Oxidation Stability

The oxidation stability analyzes of the BX samples showed that as the percentage of biodiesel increases for the blends, the shorter the induction period for the mixture. This observation is in agreement with the literature since although diesel oil is very stable to oxidation; the presence of biodiesel even at moderate concentrations (B5 and B10, for example) causes the oxidative stability of the mixture to decline significantly [37]. The Rancimat curves are shown in Figure 2.

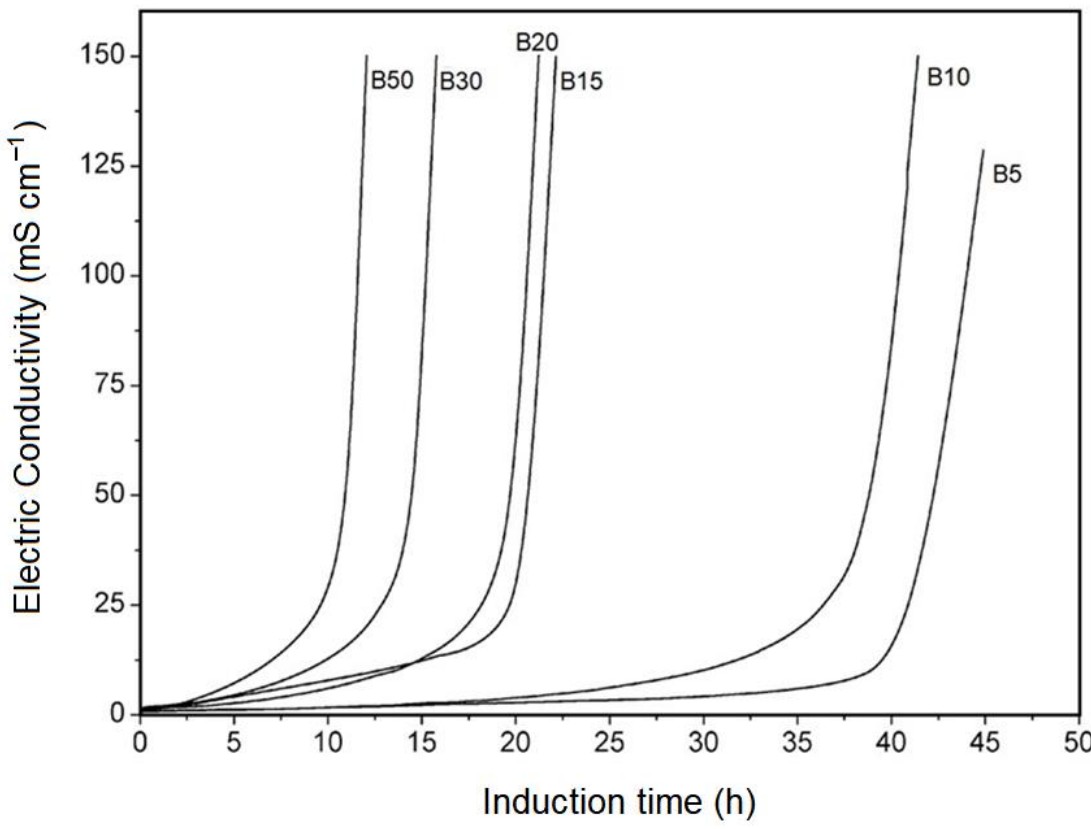

**Figure 2.** Rancimat oxidative stability curves for BX biodiesel-diesel blends.

The induction period, which is defined as the time elapsed between the start of the analysis and the time when the formation of oxidation products begins to increase rapidly, decreases with increasing biodiesel concentration, as in the blend there will be a higher concentration of esters present. These organic compounds, when subjected to an elevated temperature over a long period, will thermally degrade, forming as peroxide intermediate oxidation products, which in turn will form low molecular weight volatile organic acids, which are responsible for the oxidation of the mixture.

By observing the curves, it is clear that at a given moment there is a marked increase in the electrical conductivity of the distilled water contained in the measuring vessel. This fact indicates that there was initially the formation of peroxides, and later the formation of organic acids, indicating that the oxidation of the analyzed samples occurred. Simultaneously, it can be verified that the blends B5 and B10 presented good oxidation stability since the biodiesel present in these two mixtures is not yet in a considerable amount. However, for blends B15, B20, B30, and B50, oxidation stability has declined more sharply compared to previous blends, indicating that for blends above 10% *v/v* oxidation will occur faster.

In the results obtained from the P-DSC experiments, it was verified that the increase of biodiesel concentration in the blends decreases the oxidation stability, as previously stated from the Rancimat method. The Oxidation Onset Temperature (OOT), obtained by intersecting the extrapolation of the DSC signal relative to the exothermic oxidation event by the DSC signal baseline, decreased as the volume concentrations of biodiesel increased. The P-DSC curves are shown in Figure 3, and the data relative to the oxidation temperatures are given in Table 6.

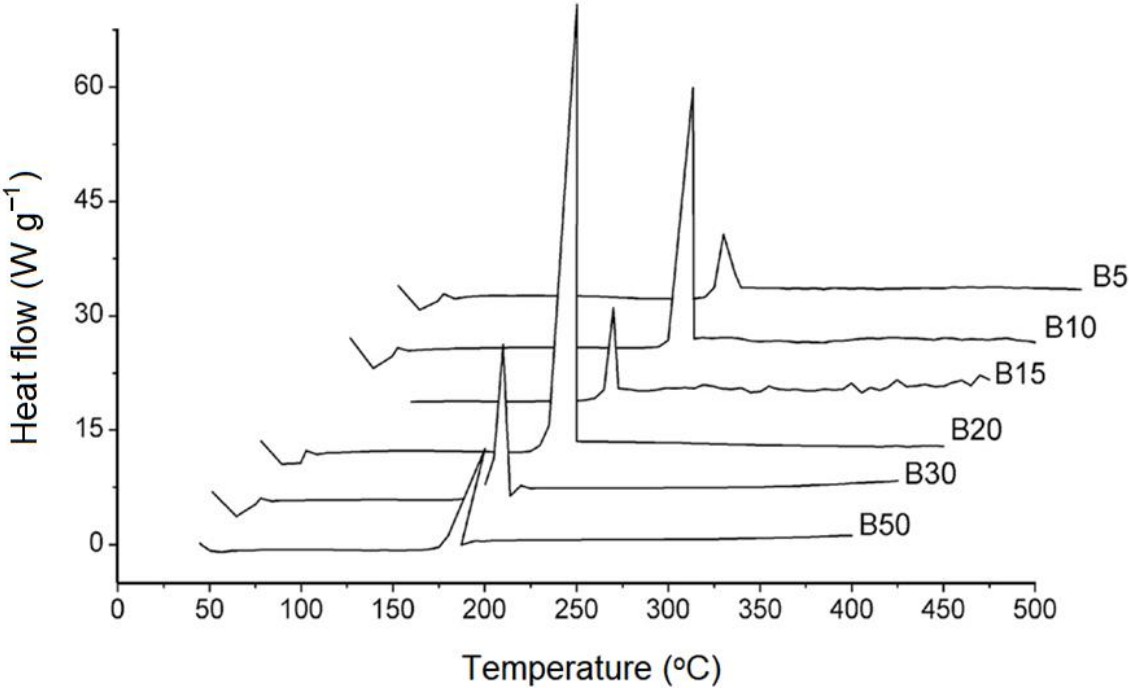

**Figure 3.** Pressurized Differential Scanning Calorimetry (P-DSC) curves for the BX biodiesel-diesel blends.

**Table 6.** P-DSC results relative to oxidation temperatures for BX blends.

| Blend | Oxidation Onset |
|:---:|:---:|
| (BX) | Temperature (°C) |
| B5 | 198.0 |
| B10 | 196.5 |
| B15 | 190.0 |
| B20 | 182.0 |
| B30 | 175.0 |
| B50 | 173.0 |

Similar to the results presented for Rancimat, for the results obtained using the P-DSC it was also verified that the blends B5 and B10 have a high initial oxidation temperature, thus being more stable than the other blends. For blends B15, B20, B30, and B50, the initial oxidation temperature decreased significantly compared to the previous blends, confirming that for blends above 10% $v/v$ volume a lower temperature will be required for the oxidation process starts under the same test conditions.

The results for the Rancimat and P-DSC tests showed that the higher the biodiesel concentration in the mixture, the lower the oxidative stability. With the analysis of the data, it is verified that there is an abrupt fall of this parameter when it exceeds the concentration of 10% to 15% $v/v$ of biodiesel. Thus, for the use of blends above 10% $v/v$ by volume of biodiesel to be viable, it is necessary to add oxidizing additives for long storage periods. Regarding the thermal properties, it was found that both blends can be used in diesel engines without any damage in terms of energy availability. Regarding the higher calorific value, the difference between blends B5 and B50 does not imply any significant loss of energy generated during combustion. Regarding the calculated cetane index, it was observed that although between the B30 and B50 blends there was a decrease in the value of this parameter, all blends have values corresponding to fuels with good ignition delay times, which implies a good functioning of the engine.

## 4. Conclusions

The obtained biodiesel/petrodiesel blends form a stable composition of the fuel. The oxidation stability of pure biodiesel is low; however, its stability was improved by the addition of petrodiesel. The mixture of biodiesel with petrodiesel would not need any addition of antioxidants and can be stored for a long period of time. For a low amount of petrodiesel in biodiesel, the blend probably would need the addition of antioxidants to resist long periods of storage. Finally, it is concluded that both blends can be used in diesel engines without further damage since no technical modifications to conventional engines are required for their use. However, it is worth noting the need for additives with oxidizing agents to mixtures above 10% $v/v$ by volume of biodiesel, so that oxidation stability is guaranteed. Regarding the ignition quality of the fuel, the obtained values of the Calculated Cetane Index for all samples was higher than 45, indicating that the obtained biodiesel/petrodiesel blends present very good ignition properties for both heavy fuel oil or marine diesel oil. Analyzing the oxidation process, biodiesel decomposes to undesired compounds and thus decreasing the fuel quality, which can form deposits in various components of the engine. The addition of petrodiesel to biodiesel can solve these problems. Thus, research on the thermal and stability of biodiesels containing petrodiesel is important for a better understanding of the blends for application as fuel.

**Author Contributions:** Conceptualization, J.S.A. and J.B.S.; methodology, R.V.B. and J.B.S.; writing—original draft preparation, G.J.T.F. and A.C.F.C.; writing—review and editing, V.J.F.J. and A.S.A. All authors have read and agreed to the published version of the manuscript.

**Funding:** This research was funded by the National Agency of Petroleum, Natural Gas, and Biofuels (ANP-Brazil).

**Acknowledgments:** The authors acknowledge the support from the National Agency of Petroleum, Natural Gas, and Biofuels (ANP-Brazil). One of us (J.S.A.) acknowledges the Coordination for the Improvement of Higher Education Personnel (CAPES-master fellowship).

**Conflicts of Interest:** The authors declare no conflict of interest.

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
