# Peer review of "Thermal Oxidative Stability of Biodiesel/Petrodiesel Blends by Pressurized Differential Scanning Calorimetry and Its Calculated Cetane Index"

_processes, doi:10.3390/pr9010174_

Round 1

Reviewer 1 Report

Tha papers gives interesting results about the test method for oxidation stability of biodiesel/diesel fuel blends. However, I think the introduction could be extended with some relevant papers on the field of investigation of oxidation stability of biodiesels. Advantages of the DSC method should be more emphasized. I feel the novelty of the results should be also highlighted more.

Author Response

Response to Reviewer 1 Comments

Point 1: The paper gives interesting results about the test method for oxidation stability of biodiesel/diesel fuel blends. However, I think the introduction could be extended with some relevant papers on the field of investigation of oxidation stability of biodiesels. Advantages of the DSC method should be more emphasized. I feel the novelty of the results should be also highlighted more.

Response 1: The introduction was improved, with relevant papers added on oxidation stability of biodiesels.

Point 2: Advantages of the DSC method should be more emphasized. I feel the novelty of the results should be also highlighted more.

Response 2: The advantage of DSC in addition of P-DSC was emphasized.

Reviewer 2 Report

The paper analyses the feasibility of diesel-biodiesel mixtures for diesel engine fuelling in terms of  thermal and oxidative properties of this fuel. The paper might have value for general practice and science, however there are some issues, which should be taken into account and incorporated into the text before publication.  

Main issue: 

The paper presents analytical tests and it has a form of a report. In its current form the paper does not present a scientific problem. I suggest to highlight scientific contribution of the paper. Please present the problem, provide a hypothesis and a proposition of its solution. Then provide an analysis and results from testing the hypothesis. 

Secondary issues: 
  1. I suggest to avoid using an abbreviation in the title. 
  2. The title should be extended as the paper presents an analysis of more parameters then it results from the current version of the title. 
  3. Keywords – Cetane Index -> Calculated Cetane Index (CCI)   
  4. The literature review should be updated – 11 references is insufficient. Please check some recent papers on biofuels published in Energies, Fuel and Processes. 
  5. In the introduction the influence of the CN/CCI on safe and reliable operation of the engine should be at least shortly presented, including its possible severe damages  (c.f. Marine Auxiliary Diesel Engine Turbocharger Damage (Explosion) Cause Analysis. Jour. of Pol. CIMAC, 2 (2), 2007, 33-40). 
  6. Line 76 - 2.1. Preparation of the BX Blends -> 2.1. Preparation of the biodiesel-diesel blends. 
  7. Line 105 – Litre - L -> dm3
  8. Section Materials and Methods – I suggest to provide the definitions for parameters presented in each subsection (2.1, 2.2. etc.). 
  9. Please add all used standards as appropriate references and add them to the list of references. 
  10. Fig. 1 – please correct the caption – provide the content of the figure. 
  11. In the whole paper: % v -> % v/v. 
  12. Line 212 – 3.8% -> 3.8% v/v. 
  13. Line 122: Cetane Index Calculated -> Calculated Cetane Index. 
  14. The final conclusion with the main results and a propositions of further research will be appreciated. 

Author Response

Response to Reviewer 2 Comments

Point 1: The paper presents analytical tests and it has a form of a report. In its current form the paper does not present a scientific problem. I suggest to highlight scientific contribution of the paper. Please present the problem, provide a hypothesis and a proposition of its solution. Then provide an analysis and results from testing the hypothesis.

Response 1: The paper was immproved, according to recommendations, as can be seen in red, pointed in the manuscript.

Point 2: I suggest to avoid using an abbreviation in the title. The title should be extended as the paper presents an analysis of more parameters then it results from the current version of the title.

Response 2: The title was changed to: “Thermal oxidative stability of Biodiesel/Petrodiesel blends by Pressurized Differential Scanning Calorimetry and its Calculated Cetane Index”

Point 3: Keywords – Cetane Index -> Calculated Cetane Index (CCI)  

Response 3: In the keywords, the Cetane Index was changed to Calculated Cetane Index (CCI)

Point 4: The literature review should be updated – 11 references is insufficient. Please check some recent papers on biofuels published in Energies, Fuel and Processes.

Response 4: The literature was updated with recent papers, including oxidative stability of biodiesel, antioxidants, Calculated Cetane Index (CCI), Rancimat and pressurized Differential Scanning Calorimetry (P-DSC).

Point 5: In the introduction the influence of the CN/CCI on safe and reliable operation of the engine should be at least shortly presented, including its possible severe damages (c.f. Marine Auxiliary Diesel Engine Turbocharger Damage (Explosion) Cause Analysis. Jour. of Pol. CIMAC, 2 (2), 2007, 33-40).

Response 5: The influence of Cetane Number and Calculated Cetane Index was included in the introduction, giving attention on the ignition quality for different heavy fuel oils or marine fuel diesel. The suggested reference was included in the paper.

Point 6: Line 76 - 2.1. Preparation of the BX Blends -> 2.1. Preparation of the biodiesel-diesel blends. Line 105 – Litre - L -> dm3.

Response 6: The new topics were corrected. We preferred describe as “Preparation of the biodiesel/petrodiesel blends. The conversion from L to dm3 was done.

Point 7: Section Materials and Methods – I suggest to provide the definitions for parameters presented in each subsection (2.1, 2.2. etc.).

Response 7: The definition of the parameters were included in the manuscript, in the section Materials and Methods.

Point 8: Please add all used standards as appropriate references and add them to the list of references.

Response 8: All standards and references were listed in the Reference section.

Point 9: Fig. 1 – please correct the caption – provide the content of the figure.

Response 9: The caption was corrected to “Distillation curves obtained according to ASTM D86 for the biodiesel/petrodiesel blends, showing the profiles for B5, B10, B15, B20, B30 and B50 samples”

Point 10: In the whole paper: % v -> % v/v.

Response 10: corrected

Point 11: Line 212 – 3.8% -> 3.8% v/v.

Response 11: corrected

Point 12: Line 122: Cetane Index Calculated -> Calculated Cetane Index.

Response 10: corrected

Point 13: The final conclusion with the main results and a propositions of further research will be appreciated.

Response 13: A conclusion was included in the manuscript.

Round 2

Reviewer 2 Report

The authors took into account all of my remarks. Thank you. The paper is much better now. I have only some minor remarks, which if taken into account by the authors might improve the quality of the paper.

  1. Line 117 please use commas instead of semicolons.
  2. Line 130 - D = Density at 15 °C in g mL-1 -> D - density at 15 °C in g/cm3 
  3. Line 154, 156 etc. - please use SI units - L (in fact should be "l") please change into dm3, ml into cm3 etc.
  4. Line 172 - (approximately 3500 kPa) -> (3.5 MPa). Without "approximately"!
  5. Table 2 and more... please unify style: vol -> v/v
  6. Table 3 - Kg -> kg
  7. Line 198 - % -> % v/v
  8. Line 255, 256, Table 5... - Please use SI units. I suggest recalculate cal into J.
  9. Fig. 2, 3... - please indicate units according to the publisher rules: / h -> (h)
  10. Line 341 - pleasu unify the style: 10%(v/v) -> 10% v/v

Author Response

Responses:

  1. Line 117 please use commas instead of semicolons. Ok
  2. Line 130 - D = Density at 15 °C in g mL-1 -> D - density at 15 °C in g/cm3. Ok
  3. Line 154, 156 etc. - please use SI units - L (in fact should be "l") please change into dm3, ml into cm3 etc. Ok
  4. Line 172 - (approximately 3500 kPa) -> (3.5 MPa). Without "approximately"! Ok
  5. Table 2 and more... please unify style: vol -> v/v. Ok
  6. Table 3 - Kg -> kg. Ok
  7. Line 198 - % -> % v/v. Ok
  8. Line 255, 256, Table 5... - Please use SI units. I suggest recalculate cal into J. Ok
  9. 2, 3... - please indicate units according to the publisher rules: / h -> (h). Ok
  10. Line 341 - pleasu unify the style: 10%(v/v) -> 10% v/v . Ok